

# Technical note: rectifying systematic underestimation of the specific energy required to evaporate water into the atmosphere

Andrew S. Kowalski[1,2]

[1]Departmento de Física Aplicada, Universidad de Granada, Granada, 18071, Spain
[2]Instituto Interuniversitario de Investigación del Sistema Tierra en Andalucía, Centro Andaluz de Medio Ambiente (IISTA -CEAMA), Granada, 18071, Spain

*Correspondence to*: Andrew S. Kowalski (andyk@ugr.es)

**Abstract.** Not all of the specific energy consumed when evaporating water into the atmosphere ($\lambda$) is due to the latent heat of vaporization ($L$). What $L$ represents is the specific energy necessary to overcome affinities among liquid water molecules, neglecting the specific work done against atmospheric pressure ($p$) when water expands in volume ($V$) from liquid to gas (pV work). Here, in the one-dimensional context typifying micrometeorology, the pV work done in such an expansion is derived based on the Stefan flow velocity at the surface boundary, yielding a simple function of the virtual temperature; additionally, an empirical formula is provided that approximates $\lambda$ quite accurately over a useful range of environmental conditions. Neglect of this pV work term has caused a systematic 3-4% underestimation of $\lambda$, and to some extent inhibited closure of the surface energy balance.





Environmental sciences have many important contexts within which the specific energy required
to evaporate water (hereinafter, $\lambda$; with units of J kg$^{-1}$) has been assumed to be uniquely due to the latent
heat of vaporization (denoted here as $L$; same units). A short list includes the surface energy imbalance
(Leuning et al., 2012), the Penman-Monteith equation (Jones, 1983), and the Bowen ratio (Euser et al.,
2014). In each case, $L$ is multiplied by the evaporative flux density $E$ (with units of kg m$^{-2}$ s$^{-1}$) to define
the latent heat flux density $LE$ (with units of W m$^{-2}$). The purpose of this note is to point out that $LE$
incompletely accounts for the energy flux density associated with evaporating water into the atmosphere
(or "evaporative energy flux density", $\lambda E$), and to derive the additional energy necessary.
Sometimes termed the "enthalpy of vaporization", $L$ represents the energy per unit water mass
required *at equilibrium* to overcome affinities among liquid-phase molecules, break their bonds and
enable the transition to the gas phase. Its empirical determination is based on the equilibrium
thermodynamics that underlie the Clausius-Clapeyron equation for the case of isothermal, isobaric phase
change (Petty, 2008). Under such conditions, $L$ is proportional to the slope of the curve relating the
equilibrium vapor pressure to temperature ($T$), and is a weak function of $T$ that has been both tabulated
(e.g., Rogers and Yau, 2009) and approximated by simple formulae (e.g., Henderson-Sellers, 1984).
Because equilibrium conditions exclude the possibility of work, an appropriate interpretation of $LE$ is the
energy flux density required for evaporation into a vacuum.
Yet environmental water evaporates into the atmosphere at a certain pressure (not a vacuum),
requiring additional energy to make space for new vapor. Environmental evaporation is not a case of
equilibrium thermodynamics, but performs work against atmospheric pressure to power the roughly
thousandfold expansion in water volume when transitioning from liquid to gas. Because the specific
volume of unevaporated liquid is negligible in comparison with that of evaporated vapor, such expansion
can be approximated as an injection process, known in fluid mechanics to require pressure/volume (pV)
work. In the micrometeorological, per-unit-area context of surface-normal ("vertical") flux densities, the
volume created represents a vertical displacement, where pV work serves to increase the gravitational
potential energy of the atmospheric column overlying the evaporative surface. The vertical velocity of
evaporation-driven Stefan flow at the atmosphere's lower boundary facilitates quantifying pV work in
relation to $E$.
The derivation begins by examining the energy flux, or power $P$ (with units of W), associated
with this upward air displacement:

$$P = \frac{dW}{dt} = \frac{Fdz}{dt} = Fw, \qquad (1)$$

where $W$ represents work (with units of J), $F$ the force corresponding to the air column weight (with units
of N), $z$ the height increment (with units of m), and $w$ the vertical velocity (with units of m s$^{-1}$). Then,
when dividing (1) by the column area $A$ (with units of m$^2$), recalling that pressure $p$ (with units of Pa) is
defined as force per unit area ($p = F/A$), and furthermore substituting for the vertical velocity of the
evaporation-induced Stefan flow ($w = E/\rho$), defined (Kowalski, 2017) by the ratio of $E$ (with units of kg
m$^{-2}$ s$^{-1}$) to the air density $\rho$ (with units of kg m$^{-3}$), the resulting energy flux density (with units of W m$^{-2}$)
can be simplified to



$$\lambda E - LE = \frac{p}{\rho} E. \qquad (2)$$

Simplifying then, with substitution from the ideal gas law
$$p = \rho R_d T_v, \qquad (3)$$

where $R_d$ is the gas constant for dry air (287 J kg$^{-1}$ K$^{-1}$) and $T_v$ the virtual temperature (with units of K),
defines the specific work (with units of J kg$^{-1}$) as
$$\lambda - L = R_d T_v. \qquad (4)$$

In short, the specific energy associated with surface evaporation is $\lambda = L + R_d T_v$ (with units of
J kg$^{-1}$), comprising both the latent heat and also the pV work – each per unit mass of water – associated
with evaporation into the atmosphere. As Table 1 shows, the systematic underestimation that has occurred
due to neglect of the pV work term (using $L$ rather than $\lambda$) is small, of order 3-4%, but hardly negligible.
Linear regression of the data in Table 1 yields a simple expression for the specific energy required to
evaporate water into the atmosphere
$$\lambda = 2579.2 - 2.023 \cdot (T - 273.15), \qquad (5)$$

which approximates $\lambda$ (with units of J kg$^{-1}$) as a linear function of $T$ (with units of K) to within +/- 0.1%
over the temperature range of Table 1 and at pressures ranging from 1100 to 600mb.
**Acknowledgements**
This research was aided by funding from the Spanish Ministry of Economy and Competitiveness project
CGL2017-83538-C3-1-R (ELEMENTAL). The author thanks Dr. R. I. Hidalgo Álvarez for insight into
equilibrium thermodynamics.

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





**Table 1 For a 1000mb pressure and a range of temperatures at which environmental evaporation occurs ($T$),**
**the specific energies of latent heat ($L$; Rogers and Yau, 2009) and total evaporation including pV work ($\lambda$),**
**along with the underestimation ($\varepsilon$, as a relative error) that has been committed when neglecting pV work.**



| $T$ (K) | $L$ (J kg$^{-1}$) | $\lambda$ (J kg$^{-1}$) | $\varepsilon$ (%) |
|---------|---------|---------|---------|
| 273.15 | 2501 | 2580 | 3.05 |
| 278.15 | 2489 | 2569 | 3.12 |
| 283.15 | 2477 | 2559 | 3.19 |
| 288.15 | 2466 | 2549 | 3.27 |
| 293.15 | 2453 | 2538 | 3.35 |
| 298.15 | 2442 | 2529 | 3.43 |
| 303.15 | 2430 | 2518 | 3.51 |
| 308.15 | 2418 | 2508 | 3.61 |
| 313.15 | 2406 | 2499 | 3.71 |
