# Peer review of "Hydrol. Earth Syst. Sci. Discuss., https://doi.org/10.5194/hess-2018-195 Manuscript under review for journal Hydrol. Earth Syst. Sci. Discussion started: 16 April 2018 © Author(s) 2018. CC BY 4.0 License."

_Hydrology and Earth System Sciences, 2018_

## Short Comment (SC1) · 18 Apr 2018

Very interesting note refining the definition of the latent heat of evaporation and its empirical dependence on temperature, of great relevance also in the eddy covariance community as related to the energy budget closure problem. I would like to point out that in my opinion the units in Tab. 1 are not correct: the L and lambda as reported are in J g-1, not in J kg-1. Eq. (5) is then also in J g-1 instead of J kg-1, unless it is multiplied by 10ˆ3. Another point is the fact that in Eq. (4) the virtual temperature is used, while in my understanding Tab.1 reports ambient temperature: I think it should be mentioned that the difference between the two is not giving relevant differences in

the calculation of lambda (if so), to improve the clarity of the note.

Many thanks to the author for this work, Best Regards

Simone Sabbatini

---

## Author Comment (AC1) · 20 Apr 2018

I thank Dr. Sabbatini very much for his positive and constructive comments. He is exactly right that the units in Table 1, and also describing equation (5) at line 70, should be J g-1. This must and will be corrected in future revision, if the editor allows.

The differences between the temperature and virtual temperature are not important. The latter is used in the ideal gas law (equation 3) to avoid the need for a variable "constant" (the particular gas constant for moist air), following meteorological tradition. It was furthermore used, as in equation (4) to determine the values of lambda that are tabulated in Table 1. By contrast, equation (5) is merely an empirical relation to

approximate the temperature dependence of lambda, determined by regression from the data in Table 1. The temperature was chosen to express this empirical dependence, both for simplicity and because it approximates lambda to within +/- 0.1%. To make this clearer, I propose to add text to the legend for Table 1, indicating that the values of lambda were calculated using equation (4).

―――――――――――――――――

---

## Short Comment (SC2) · 22 Apr 2018

In the abstract of this paper, we read:

"What $L$ represents is the specific energy necessary to overcome affinities among liquid water molecules, neglecting the specific work done against atmospheric pressure (p) when water expands in volume ($V$) from liquid to gas ($pV$ work)."

In the body of the paper, we then read:

"[$L$'s] empirical determination is based on the equilibrium thermodynamics that underlie the Clausius–Clapeyron equation for the case of isothermal, isobaric phase change

(Petty, 2008)."

But equation (7.5) of Petty (2008) explicitly shows $L$ as consisting of both an internal energy part (the energy of attraction of the molecules) and a pressure-volume part. So there seems to be an inconsistency between the premise of the present paper and the definition of $L$ as cited, according to which $L$ is an enthalpy just like sensible heat (as it should be).

———————————————

---

## Short Comment (SC3) · 22 Apr 2018

It appears that underlying the author's premise is the notion that pressure-volume work due to expansion against *atmospheric* pressure must be accounted for (e.g., line 37, "Environmental evaporation is not a case of equilibrium thermodynamics, but performs work against atmospheric pressure."). But bulk mechanical expansion of evaporating water against full atmospheric pressure occurs only with actual boiling, at which point atmospheric and vapor pressures are by definition the same.

For sub-boiling temperatures, evaporation is necessarily a diffusive process rather than a bulk mechanical expansion. In molecular diffusive processes, I believe it is widely

accepted that, in the absence of attractive forces between air molecules and vapor molecules, the presence or absence of passive air molecules plays no role in the thermodynamic equilibrium between the liquid and vapor phases of water or in the energy of the phase change. Thus, both the saturation vapor pressure $e_s(T)$ and the latent heat (or enthalpy) of vaporization $L$ are the same whether or not air molecules are present, and the pressure-volume component of $L$ is always the volume change (due to vapor alone) times the saturation vapor pressure at that temperature.

---

## Author Comment (AC2) · 24 Apr 2018

I thank Professor Petty for engaging on this issue. I proposed this referee because of the combination of his authority on the matter and doubts regarding my derivation. I believe that he is correct and will withdraw the manuscript from consideration for publication.

---

## Author Comment (AC3) · 24 Apr 2018

Although I will withdraw the manuscript based on Professor Petty's first comment (hess-2018-195-SC2), I must push back somewhat regarding this second comment (hess-2018-195-SC3), with which I partially disagree.

First let me note where we agree. Boiling does indeed represent a case of pure bulk mechanical expansion, which I might have termed "non-diffusive" transport (Kowalski, 2017). Since the atmospheric and vapor pressures are the same, according to Dalton's law there is no other gas species present near the boiling surface, and therefore no diffusion.

[Figure]

However, for sub-boiling temperatures the relevance of diffusion does not put a stop to bulk mechanical expansion. Rather, according to equations (7) and (8) of Kowalski (2017), the water vapor mass fraction (or specific humidity, q) represents the non-diffusive fraction of vapor transport, while the balance (1- q) is the diffusive fraction. Thus, instead of "evaporation is necessarily a diffusive process rather than a bulk mechanical expansion", I would say that "evaporation is a diffusive process *in addition to* a bulk mechanical expansion", where the two processes' degrees of relevance depend on q.

This difference of opinion regards transport alone, and does not apply to Dr. Petty's conclusion regarding thermodynamics. I accept that neither the saturation vapour pressure nor the latent heat of vaporization depend on the pressure of dry air.

---

## Referee Comment (RC1) · Anonymous Referee #1 · 12 May 2018

Because the author has agreed to withdraw the paper, there seems little benefit or need for me to comment. Nonetheless, as a reviewer I have been asked to do so. Dr Petty's point about the work done by the expansion of a gas during evaporation being part of the enthalpy of vaporization is correct.

The enthalpy of vaporization, $L$, represents a change in the enthalpy of the system, $\Delta H$, and all the thermodynamics texts that I am familiar with state that the first law of thermodynamics during an evaporative process is expressed as $L = \Delta H = \Delta U + p\Delta V$, where $\Delta U$ represents the change in internal energy of the system and $p\Delta V$ is the work done by expansion. Consequently, the author's premise – as stated in the first two

sentences of his Abstract: "Not all of the specific energy consumed when evaporating water into the atmosphere ($\lambda$) is due to the latent heat of vaporization (L). What L represents is the specific energy necessary to overcome affinities among liquid water molecules, neglecting the specific work done against atmospheric pressure (p) when water expands in volume (V) from liquid to gas (pV work)." – is false. I would not recommend this paper for publication.

---

## Referee Comment (RC2) · A.G.C.A. Meesters (Referee) · 18 May 2018

First, I fully agree with the comments that the $L$ used in standard practice is an "enthalpy", which means that it already includes the work done by expansion against the air pressure when the water evaporates. So the point made by the Note is incorrect.

At the same time, I must say that the Note does not come as a great surprise to me ! I was wondering about the matter years ago, and fortunately I found the correct answer in books on thermodynamics. But since then I have paid attention to the matter, and I have seen that some texts define latent heat without hesitation in term like "energy needed to overcome intermolecular forces which hold the substance together as a liquid" etcetera; just like the definition that is the starting point of the Note. And once a reader is prejudiced by this, he/she is not readily corrected even by the some of the most respectable textbooks of hydrology and meteorology, whose definitions ("energy needed to evaporate water"), are quite ambiguous in this respect. One time I also encountered an internet discussion on the subject, and it went wrong. Added to this, the well-known problems with not-closing energy balances make it tempting to doubt everything in the end, including latent heat values. Hopefully, the note and the surrounding discussion may be of some use after all.

For completeness, something should be added about the magnitude of the work-term, which is a thing of interest for its own sake. In the Note, the Stefan flow velocity $w$ is expressed as $w = E/\rho$ with $E$ the evaporation and $\rho$ the mass density of the *air*. But it would seem that consistency requires instead the use of the mass density of *water vapor*, put in equilibrium with the air pressure according to the gas law:

$$\rho = \frac{p}{R_w T}$$

in which $R_w$ is the gas constant for water. If this is substituted, the calculation in the Note would end up with

$$\lambda - L = R_w T$$

and not $R_d T_v$ as stated in the Note.

This value is the classical result obtained in the thermodynamics textbooks experiments where water evaporates into an empty cylinder (no air inside), topped by a piston. It turns out that its use is allowed irrespective whether the vapor is mixed with air molecules or not, see the end of G. Petty's second comment (SC3). In accordance with this, it is of old the practice of hydrology and meteorology to use latent heat values which are identical to the ones obtained by laboratory physics.

[Figure]

It would be most helpful if someone could give a reference to literature where the whole matter (including the open air case) is explained in a clear way. I could not find one, possibly one will have to search in the literature of an older era, which is a bit less accessible nowadays.

---

## Author Comment (AC4) · 13 Jun 2018

Like Anonymous Referee #1, I am required by the journal to reply to his/her comment. I have nothing further to say except that the paper will be withdrawn.

---

## Author Comment (AC5) · 13 Jun 2018

I thank Antoon Meesters for his reflections on this matter. The paper will be withdrawn.